# The Interrelated Multifactorial Actions of Cortisol and Klotho: Potential Implications in the Pathogenesis of Parkinson’s Disease

**DOI:** 10.3390/brainsci12121695

**Published:** 2022-12-10

**Authors:** Nijee S. Luthra, Angela Clow, Daniel M. Corcos

**Affiliations:** 1Department of Neurology, University of California San Francisco, San Francisco, CA 94127, USA; 2Department of Psychology, School of Social Sciences, University of Westminster, London W1B 2HW, UK; 3Department of Physical Therapy & Human Movement Sciences, Feinberg School of Medicine, Northwestern University, Chicago, IL 60208, USA

**Keywords:** cortisol, klotho, Parkinson’s disease, aging, stress

## Abstract

The pathogenesis of Parkinson’s disease (PD) is complex, multilayered, and not fully understood, resulting in a lack of effective disease-modifying treatments for this prevalent neurodegenerative condition. Symptoms of PD are heterogenous, including motor impairment as well as non-motor symptoms such as depression, cognitive impairment, and circadian disruption. Aging and stress are important risk factors for PD, leading us to explore pathways that may either accelerate or protect against cellular aging and the detrimental effects of stress. Cortisol is a much-studied hormone that can disrupt mitochondrial function and increase oxidative stress and neuroinflammation, which are recognized as key underlying disease mechanisms in PD. The more recently discovered klotho protein, considered a general aging-suppressor, has a similarly wide range of actions but in the opposite direction to cortisol: promoting mitochondrial function while reducing oxidative stress and inflammation. Both hormones also converge on pathways of vitamin D metabolism and insulin resistance, also implicated to play a role in PD. Interestingly, aging, stress and PD associate with an increase in cortisol and decrease in klotho, while physical exercise and certain genetic variations lead to a decrease in cortisol response and increased klotho. Here, we review the interrelated opposite actions of cortisol and klotho in the pathogenesis of PD. Together they impact powerful and divergent mechanisms that may go on to influence PD-related symptoms. Better understanding of these hormones in PD would facilitate the design of effective interventions that can simultaneously impact the multiple systems involved in the pathogenesis of PD.

## 1. Introduction

Parkinson’s disease (PD) is the fastest growing neurological disorder globally [1,2]. Fueled by aging, the number of people with PD is expected to exceed 12 million by 2040 [1]. PD is characterized as a movement disorder, resultant from progressive neurodegeneration of dopamine neurons in the substantia nigra pars compacta and its projections. However, it is increasingly apparent that other brain regions are affected, e.g., decreased metabolism in the prefrontal, occipital, and parietal cortices as well as changes in the lentiform nucleus, thalamus, pons and cerebellum [3]. Dopaminergic and non-dopaminergic pathology contribute to a wide range of non-motor symptoms (NMS), including cognitive impairment, mood disorder, circadian disruption, autonomic dysfunction, fatigue, and apathy [4,5,6]. Despite the heterogeneity in pathogenesis and symptomatology, aging remains the single most important risk factor for development of PD and of NMSs that disrupt the quality of life [7,8,9].

Mechanisms of aging and neurodegeneration are thought to be interrelated with chronic stress. Both aging and chronic stress are pertinent to the pathogenesis of PD and impact mitochondrial dysfunction, oxidative stress, inflammation, and changes in metabolism [10]. These pathways result in changes in cellular function such as decreased ATP synthesis, increase production of reactive oxygen species (ROS) and Ca+ accumulation, microglia response, and increased proinflammatory cytokines and infiltrating immune cells, all contributing to cell apoptosis. These processes occur in vulnerable brain regions and interplay with genetic and environmental risk factors to contribute to the development and progression of PD [8,10,11]. Here, we review the actions of two hormones known to be implicated in the aging and stress processes and discuss their potential roles in PD pathophysiology. Cortisol is a critical hormone involved in normal stress responsivity, physiological homeostasis, and circadian function. Excess secretion is associated with a wide range of maladaptive correlates of aging and chronic stress [12,13,14,15]. Relatively newly discovered, klotho is a longevity hormone that delays aging and enhances cognition [16,17,18]. In this review, we highlight the yin-yang roles of cortisol and klotho in key aging and stress pathways and how this may associate with progression and symptoms of PD.

## 2. Function of Cortisol and Klotho

### 2.1. Cortisol

Cortisol secretion is the product of hypothalamic-pituitary-adrenal (HPA) axis activation. It is a powerful steroid hormone that can pass into every cell of the body, with genomic and non-genomic actions affecting mitochondrial, immune, and metabolic function [19]. The brain is a prominent target for cortisol and thus a central structure for adaptation to stress. A wide brain network involving the hippocampus, amygdala, prefrontal cortex and brainstem nuclei are involved in HPA axis activation in response to acute or chronic stress [20]. The underlying basal secretory activity of the HPA axis is regulated by the hypothalamic central pacemaker: the suprachiasmatic nucleus (SCN). The SCN transmits its circadian signal to peripheral clock genes via neural and hormonal mechanisms, with cortisol secretion playing a significant role, synchronizing circadian oscillations throughout the body [21].

Cortisol has pleiotropic effects on the brain, affecting mood, behavior, cognition, and programming of the stress response [22]. Cortisol is necessary for neuronal differentiation, integrity, and growth, as well as synaptic and dendritic plasticity [23,24]. These processes in turn support brain functions such as decision-making, reward-based behavior, motor control, visual information processing, learning and memory, and energy regulation. Cortisol actions are mediated by the glucocorticoid receptors (GRs) and mineralocorticoid receptors (MRs). In addition to rapid non-genomic mechanisms, cortisol influences brain functions by activating GR-mediated gene transcription [25]. Some of these target genes code for neurotrophic factors and their receptors, anti- and pro-inflammatory markers, signal transduction, neurotransmitter catabolism, energy metabolism, and cell adhesion [25,26,27]. Stress-induced shifts in cortisol associate with time- and region-dependent changes in neuronal activity to promote the brain’s adaptation to the continuously changing environment in the short and long-term [28]. A dysfunctional HPA axis is thought to occur in PD, leading to GR desensitization and high circulating cortisol concentrations [29].

### 2.2. Klotho

*KLOTHO* (*KL*) is a serendipitously discovered gene on chromosome 13 found to have profound effects on lifespan [30]. Deficiency of klotho protein in mice severely shortens lifespan and prompts signs of premature aging while overexpression of klotho increases lifespan by ~30%. Since its initial discovery, increased levels of klotho have been associated with longevity in several populations and decreased levels of klotho have been associated with aging-related diseases including cancer, cardiovascular disease, kidney disease, and recently neurodegenerative diseases [31,32,33,34,35,36,37]. 

Klotho is expressed primarily in the kidneys and choroid plexus [30,38,39]. Klotho mRNA is also detectible in many other brain regions, including cortex, hippocampus, cerebellum, striatum, substantia nigra, olfactory bulb, and medulla [38,39,40,41]. Klotho’s actions are complex and multidimensional—it suppresses insulin and *Wnt* signaling [42,43], regulates ion channel clustering and transport [44], modulates *N*-methyl-d-aspartate receptor (NMDAR) signaling [18] and promotes fibroblast growth factor (FGF) function [45]. Klotho, linked with FGF23, is important for regulation of calcium, phosphate, and vitamin D homeostasis [46,47,48,49,50,51]. Klotho linked with FGF21 is involved in stimulation of the starvation response, activating the HPA axis and sympathetic nervous system, as well as increasing intracellular klotho expression in the SCN within the hypothalamus [52].

## 3. Cortisol and Klotho in Neurodegenerative Disease

Cortisol and klotho have relevance to various neuropsychiatric and neurodegenerative disorders. Cortisol levels are increased in individuals with depression, sleep disturbances, and neurodegenerative diseases like AD and PD [53,54,55,56,57,58]. Conversely, lower circulating klotho levels are reported in bipolar disorder, depression, multiple sclerosis, temporal lobe epilepsy, AD and recently, PD [36,37,59,60,61,62].

Accumulating evidence suggests that the HPA axis is dysregulated in PD. Cortisol levels are found to be increased in toxin animal models of PD [63]. Elevated cortisol levels also induce impairment of motor function and accelerate nigral neuronal loss in rats exposed to chronic stress and subsequent increase in cortisol [64]. Individuals with PD have elevated cortisol secretion in blood and saliva, especially in the morning [58,65,66]. More recently, glucocorticoid concentrations measured in the hair of PD patients showed an excess of cortisone, the main cortisol metabolite, but not cortisol itself [67]. 

Klotho was initially linked to neurodegenerative diseases in studies of AD. Klotho is decreased in the CSF of AD patients [36]. Higher klotho is also associated with reduced amyloid-beta (Aβ) burden and improved cognition in populations at risk for AD [68]. Emerging studies are now connecting klotho with PD. Klotho-insufficient mice develop neurodegeneration of mesencephalic dopaminergic neurons in substantia nigra and ventral tegmentum area, while klotho overexpression protects dopaminergic neurons against oxidative injury [40,69,70]. Exogenous klotho administration demonstrates neuroprotective potential in toxin rat models of PD through alleviation of astrogliosis, apoptosis, and oxidative stress [71]. One study reported that while plasma klotho levels were not significantly different between people with PD and healthy controls, klotho levels were lower in men compared to women with PD [72]. This is interesting since sex is an important biological factor in development and phenotype of PD as well as hormonal regulation in general. Another study looking at two independent cohorts of people with PD found that CSF levels of klotho were lower in people with PD compared to healthy controls [37]. A recent perspective by Grillo et al. (2022) also points out that enteric cells express klotho, and both blood and enteric levels of klotho are altered in the setting of gut disease or inflammation [73]. The authors go on to suggest that since PD pathology is hypothesized to start in the enteric nervous system, this poses an important need to assess klotho in the gastrointestinal tract of people with PD and evaluate whether modulation of klotho in the gut may serve as a disease-modifying strategy. 

Lastly, it is important to note how the main PD symptomatic treatment, levodopa, is associated with cortisol and klotho. Administration of levodopa decreases HPA axis activity, thereby decreasing cortisol levels [74,75]. It is currently unclear if levodopa affects klotho levels or vice versa and future studies should take this into account.

## 4. Factors Modulating Cortisol and Klotho Regulation

### 4.1. Aging

Aging is the most critical risk factor for PD [7,76], yet the relationship between molecular/cellular processes of healthy aging and those of PD pathogenesis remain unclear. It can be hypothesized that specific regions of the PD brain (e.g., dopaminergic neurons in substantia nigra pars compacta) undergo localized, accelerated aging [77]. Aging links together several pathological mechanisms known to play a significant role in PD—from increased inflammation and oxidative stress to mitochondrial dysfunction and dysregulation of lysosomal, proteasomal and autophagic functions– and all likely to contribute to neurodegeneration. The concept of “inflammaging” has been proposed as a principal mechanism in PD [78] and describes the sustained systemic inflammatory state that develops with advanced age [79]. This chronic inflammation is thought to result from exposure to chronic stressors and/or imbalance between inflammatory and anti-inflammatory networks. 

The convergence of cortisol and klotho along the pathways of aging has notable implications for PD. Cortisol levels decrease in decades 20s–30s, are relatively stable in 40-50s, and increase after age 60 [80]. Elevated cortisol has been reported in age-related illnesses such as cardiovascular disease, type II diabetes mellitus, osteoporosis, and cognitive impairment [81,82,83,84]. In contrast, klotho levels are highest at birth in humans, with levels 7-fold higher than adulthood levels, and decline after age 40 [85,86]. Several studies link klotho to increased lifespan and better health outcomes, including decreased risk for cardiovascular disease and stroke, decreased macrovascular complications in patients with type 2 diabetes, and improved grip strength [33,86,87]. Klotho has also recently been included in a panel of biomarkers that may predict frailty in the elderly [88].

### 4.2. Stress

The notion that chronic stress, in addition to aging, may play a role in the pathogenesis of PD has been controversial over the years but, despite some inconsistencies in the literature, is now generally recognized [29]. In one population-based cohort study of over 2 million males, higher job demands and expectations increased PD risk [89]. In another study, the risk of PD significantly increased with the number of exposures to stressful events [90]. Post-traumatic stress disorder and adjustment disorder, both indicating occurrence of significant stressors, also associated with increased risk of PD, independent of comorbid depression or anxiety [91,92]. 

Stress is known to affect functions of the limbic system such as learning, memory and emotions [93]. The hippocampus has extensive distribution of GRs and plays a crucial role in the biological effects of chronic stress [94,95]. Stress and hypercortisolemia also disrupt sleep [96], which exerts powerful effects on the hippocampus and affects initial learning and memory consolidation [97]. Recent evidence shows that stress also modulates motor system function [98]. Since most parts of the motor system express GRs, their circuits are susceptible to the influence of cortisol. Stress can modulate movement through activation of the HPA axis and via stress-associated emotional changes. In PD mouse models, chronic stress exposure worsens motor deficits, aggravates the neurodegeneration of the nigrostriatal system, and completely blocks compensatory recovery of motor tasks [64]. A recent viewpoint by van der Heide et al. (2020) proposes a model of how chronic stress in patients with PD, resulting in higher cortisol levels, can lead to both higher susceptibility for depressive and anxiety disorders and a more rapid progression of the disease [99]. The authors review evidence on how chronic stress reduces levels of brain derived neurotrophic factor (BDNF), inducing atrophy in key brain regions of mood and behavior, and creates a proinflammatory environment that increases nigrostriatal cell loss. 

Given klotho’s role in healthy aging, it is no wonder that it too has been linked to stress. Klotho levels are reported to be lower in caregivers with chronic high stress and show an age-related decline [60]. Klotho genetic variations that alter klotho levels influence the effects of stress on cellular aging, as evidenced by changes in multiple biomarkers of aging, including telomere length, CRP levels, metabolic dysfunction and white matter microstructural integrity [100]. Mice with chronic stress demonstrate downregulation of klotho in the nucleus accumbens (NAc) and depressive-like behavior [101], responses modulated by Klotho regulation of NMDARs. 

### 4.3. Genetics

PD pathogenesis is mediated by an interaction between multiple environmental and genetic factors. The role of cortisol and klotho in PD may also be dependent on genetic variations that dictate levels or function of these two hormones. 

As discussed previously, cortisol functions by binding to GRs and MRs. Genetic variation in GR has been postulated to play a role in the physiological response to endogenous cortisol. Over 3000 single nucleotide polymorphisms (SNPs) in the GR gene have been documented. Most studies show that *BclI* and N363S gene variants are associated with clinical measures of increased glucocorticoid sensitivity, while the ER22/23EK and GR-9β are associated with decreased glucocorticoid sensitivity [102]. The ER22/23EK polymorphism links with improved survival and lower levels of the inflammatory marker C-reactive protein (CRP) [103]. *Bcl*I has consistently been shown to be associated with a higher susceptibility to major depression [104]. It is currently unclear if and how these genotypes impact PD. Lastly, polymorphisms in the catechol-O-methyl-transferase (COMT) gene have also been associated with glucocorticoid responsivity and cortisol levels. In particular, individuals with the Met/Met COMT homozygote polymorphisms are more sensitive to stressful events and have higher cortisol responses [105]. Interestingly in people with PD, Met/Met COMT homozygote polymorphism associates with lower IQ score and greater motor severity of disease [106].

Genetic variations in the *KL* gene may influence systemic klotho levels or its function. *KL* variant *rs9315202* downregulates klotho mRNA expression and associates with advanced epigenetic age and elevated aging markers such as CRP [100,107]. There is also a well-studied protective variant, termed *KL-*VS, that contains two SNPs, rs95536314 and rs9527025, in complete linkage disequilibrium. Carrying one copy of *KL*-VS increases klotho levels [18,108], while carrying two copies, decreases it [108]. *KL*-VS heterozygosity is associated with longer lifespan [31], slowed epigenetic age [100], and higher cognitive function [18,108] in most but not all populations. In a population at risk for dementia, *KL*-VS allele attenuates Aβ burden and associates with reduced risk of conversion to mild cognitive impairment or AD [68]. In PD, *KL*-VS heterozygotes have higher CSF klotho levels; however, the haplotype itself is associated with shorter interval between onset of PD and progression to MCI and worse motor phenotype [37]. Therefore, it remains to be clarified how genetic variations in *KL* gene may affect function of the klotho protein and affect individuals with PD.

It is unknown whether there are direct correlations between cortisol or klotho and the genes linked to PD. However, as both hormones are involved in mechanisms that become aberrant in PD, there may be undiscovered connections. For example, both cortisol and klotho influence mitochondrial function in opposite ways and may interact with genetic variations in genes linked to mitochondrial dysfunction in PD (i.e., Parkin, PINK1, DJ-1, LRRK2).

### 4.4. Physical Exercise

Just as cortisol and klotho are sensitive to aging and psychological stress, they are also responsive (in the opposite direction) to external stimuli that promote healthy aging, such as physical activity. The past decade has produced much evidence to support that physical exercise is potentially neuroprotective in PD. 

Physical exercise can improve dysregulated cortisol levels in healthy individuals and those with major depressive disorder [109]. High intensity exercise (>90% heart rate reserve) decreases fluctuations in salivary cortisol [110]. Another study showed that exercising intensely (70% heart rate reserve) suppresses the subsequent cortisol response to a psychosocial stressor [111]. Smyth et al. have shown that high intensity treadmill exercise in individuals with PD reduces cortisol secretion during the post-awakening period after nocturnal sleep [112]. Yoga and dance movement therapy also lead to decreased cortisol levels [113,114].

Contrarily, klotho levels are amplified by treadmill exercise in both young and aged mice [115,116]. In humans, higher levels of klotho are associated with superior lower extremity strength and functioning [117,118] Klotho levels also tend to be higher in exercise-trained individuals compared to their untrained counter-partners [119]. In healthy adults, various exercise programs (endurance, resistance, high intensity interval training) boost plasma klotho levels either acutely or after a 12–16-week training period [120,121,122,123,124]. Recent study also finds that yoga, consisting of deep breathing exercises, meditation, and postures, upregulates expression of the *KL* gene [125]. 

## 5. Candidate Mechanisms of Cortisol and Klotho Interactions in PD

Together cortisol and klotho represent powerful yet complementary mechanisms by which life stress can be internalized and aging can be regulated. We go on to highlight 4 candidate mechanisms where klotho and cortisol may be competing in the life course of PD. 

### 5.1. Mitochondrial Dysfunction and Oxidative Stress

Mitochondrial dysfunction (altered morphology, turnover and transport) plays a fundamental role in the pathogenesis of PD by chronic production of reactive oxygen species and induction of α-synuclein misfolding, promoting neurodegeneration in the substantia nigra [126]. The causes of mitochondrial dysfunction are complex and multipart, including damage to mitochondrial DNA, environmental neurotoxins, and mutations of the PINK1, DJ-1, and Parkin genes linked to PD [127,128]. In addition, mitochondrial function is intimately interlinked with other cell processes including iron, copper, and glutathione metabolism [129]. Dysfunction in any one process impacts the others, leading to a disruptive vicious cycle driving neuronal cell death and pathology. Mitochondrial dysfunction also occurs as a consequence of aging [130] and chronic stress [131,132] with changes linked to increased inflammatory responses. This is because, in addition to their other roles, mitochondria are now considered central hubs in regulating innate immunity and inflammatory responses [133]. 

The hormone cortisol is intricately coupled with mitochondrial function. Once the HPA axis is activated, it is synthesized within the mitochondria of the zona fasciculata of the adrenal cortex and has potent reciprocal effects on mitochondrial function throughout the body [134]. In this way, mitochondria are both mediators and targets of the main stress axis, with cortisol as a liaison for whole body mitochondria-to-mitochondria communication to regulate energy metabolism. Aging and stress-associated increase in cortisol can reduce the activity of specific mitochondrial electron transport chain complexes and increase mitochondrial oxidative stress [134]. In mouse models of PD, psychological stress diminishes up to 50% of mitochondrial respiration and glycolysis and links to cell death and exacerbation of motor symptoms [135]. 

Age-related declines in klotho can drive dysfunctional mitochondrial bioenergetics in skeletal muscle and kidney whilst systemic delivery of exogenous klotho rejuvenates and enhances function [136,137]. Mitochondria not only regulate the normal ROS level, but excessive ROS can also directly damage mitochondria and lead to apoptosis and cell death. Klotho induces the expression of the manganese superoxide dismutase (MnSOD) protein, a mitochondrial antioxidant enzyme that detoxifies superoxides, and thus reduces ROS [138]. A study in human stem cells shows that klotho attenuates cellular damage and cell apoptosis induced by oxidative stress by protecting mitochondrial structure [139]. Klotho is also known to play a significant role in brain metabolism as an antioxidant [140,141]. Reduction at these levels result in the inability of astrocytes to rapidly modify their metabolic activity to support adjacent neurons, making them more vulnerable to neurodegeneration [142]. 

### 5.2. Neuroinflammation

Several lines of evidence from humans and animal models support the involvement of inflammation in the onset and progression of PD. While inflammation may be a consequence of neuronal loss in PD, the chronic inflammatory response may also contribute to the progression of PD. Neuroinflammation stems from crosstalk between neurons, microglia, astroglia and endothelial cells [133], which are susceptible to α-synuclein aggregates and mitochondrial dysfunction. Under disease conditions, the homeostatic functions of the microglia and astroglia are disrupted, leading to reduced secretion of neurotrophic factors, increased secretion of proinflammatory cytokines (interleukin (IL)-6, IL1β, Tumor Necrosis Factor α (TNFα), interferon (IFN)-γ, etc.)) and chemokines (CCL2, CXCL1, etc.) and increased receptor expression for proinflammatory markers and major histocompatibility complex (MHC-I) in microglial cells [143]. Additionally, peripheral immune cells (such as CD4+ T cells) are recruited to the brain parenchyma, further augmenting the proinflammatory environment.

Cortisol plays a significant and beneficial role in regulation of inflammation, but old age and chronic stress are associated with dysregulation of the HPA axis and cortisol secretion [144]. Cortisol and pro-inflammatory cytokines interact on multiple levels. Under normal conditions, cortisol inhibits the immune system cells that produce peripheral cytokines. It also inhibits transcription and action of many of the pro-inflammatory cytokines including IL-1β, IL-6, and TNFα [145,146,147]. In a reciprocal relationship, cytokines can also influence glucocorticoid secretion, availability, and signaling. IL-1β and IL-6 can activate the HPA axis directly, while IL-1 and TNFα can impair cortisol signaling by interfering with GR phosphorylation [148,149]. In settings of chronic stress, excessive cortisol secretion leads to compensatory down-regulation or resistance of the GR and its anti-inflammatory actions, resulting instead in a pro-inflammatory milieu facilitating a wide range of disease risks including neurotoxicity [12,13]. In this scenario, increased levels of cortisol are associated with an increase in pro-inflammatory cytokines such as IL-6 [150]. Therefore, cortisol can have dual effects—it can limit inflammation under normal conditions but promote inflammation under conditions of chronic stress.

Recent studies suggest that klotho could also play a role in mediating the interface between the brain and immune system in the choroid plexus. Selectively reducing klotho within the choroid plexus of mice triggers inflammation and enhances activation of innate immune cells [151]. As a separate pathway, klotho suppresses activation of macrophages by enhancing FGF23 signaling [151]. Klotho also decreases activation of NF-κB and influences expression of the pro-inflammatory cytokines, IFNγ and TNFα [152,153], the latter being a ‘master regulator’ of production of pro-inflammatory cytokines. To counter inflammation, klotho increases production of IL-10, which is responsible for inhibiting the expression of pro-inflammatory cytokines such as TNFα [154]. 

A key event in the neuroinflammatory processes is the activation of inflammasomes, multiprotein complexes that mediate pro-inflammatory cytokine secretion and maturation. The inflammasome component NLRP3 is strongly linked to neuroinflammation. Klotho overexpression inhibits the NLRP3/caspase signaling pathways and enhances cognition in animal models of neurodegenerative disease [155]. In contrast, high cortisol levels activate NLRP1 and NLRP3 inflammasomes and promote neuroinflammation and neuronal injury [156,157].

### 5.3. Insulin Resistance

Insulin and insulin-like growth factor 1 (IGF-1) signaling represent an evolutionary conserved pathway of longevity. Oxidative stress is implicated in the onset and progression of insulin resistance and type 2 diabetes [158], which is a prominent feature of normal aging and a preclinical indicator in many neurodegenerative disorders [159]. Peripheral insulin resistance is related to impairment of central insulin signaling and reported to be an early etiological factor in development of PD [160,161]. It is also associated with more rapid progression of PD disease and related cognitive impairment and dementia [162]. Functional brain imaging in PD further shows hypometabolism in the inferior parietal cortex and the caudate nucleus, which correlate with cognitive deficits and motor symptoms, respectively [163]. In PD, insulin resistance is proposed to lead to a state of bioenergetic failure and hypometabolism in the brain that may promote neurotoxicity [164]. 

Elevated cortisol is a major causal candidate for the development of insulin resistance with aging. It is well-known that while insulin exerts anabolic actions, cortisol exerts catabolic actions and the two hormones counteract each other in many metabolic functions, from glucose utilization to lipid storage [165]. On the other hand, klotho deficiency decreases insulin production and increases insulin sensitivity [141,166]. While the mechanisms of this are not entirely clear, it is known that klotho suppresses the downstream signaling pathway of the IGF-1 reception and insulin receptor substrate without directly binding to these receptors. Insulin also increases shedding of klotho, thereby increasing circulating klotho [167]. 

### 5.4. Vitamin D Metabolism

Another route by which cortisol and klotho may contribute to the multifactorial toxic cycle implicated in PD is via their actions on the neuroprotective hormone vitamin D. Vitamin D is a fat-soluble hormone that can pass the blood–brain barrier, supporting its significance in the central nervous system. Vitamin D insufficiency is associated with an increased risk of several CNS diseases including PD [168,169]. Vitamin D is reported to regulate more than 200 genes, influencing a variety of cellular processes such as neurotransmission neuroprotection, and downregulation of inflammation and oxidative stress [170]. It stimulates expression of many neurotrophic factors including neurotrophin 3 (NT-3), BDNF, glial cell-derived neurotrophic factor (GDNF), ciliary neurotrophic factor (CNTF), and neuroprotective cytokine IL-34 [171]. 

The vitamin D receptor is found throughout the human brain but crucially, is abundant in the substantia nigra pars compacta, the primary target of neurodegeneration in PD [172]. Additionally, 1α-hydroxylase—the enzyme that converts vitamin D to its active form,1,25(OH)_2_D_3_—is highly expressed in the substantia nigra, suggesting that vitamin D may be directly or indirectly related to the pathogenesis of PD via loss of protection for vulnerable dopaminergic neurons in this brain region [173]. In the past two decades, a high prevalence of vitamin D deficiency has been noted in individuals with PD [174]. Vitamin D concentrations also negatively correlate with PD risk and disease severity [175]. A small but significant association between vitamin D status at baseline and disease motor severity at 36 months has been reported [176]. Higher vitamin D concentrations link to better cognitive function and mood in individuals with PD [170,177]. Unfortunately, a (somewhat limited) trial of Vitamin D supplementation did not appear to improve PD symptoms [178]. 

Vitamin D is now suggested to be a biomarker of healthy aging with a strong association between low levels and higher all-cause mortality with large and significant effect sizes in multiple studies [179]. Cortisol has an antagonist relationship with vitamin D. Higher cortisol levels correlate with lower vitamin D levels [180]. Several studies also suggest that vitamin D may regulate the HPA axis. In hippocampal cell cultures, vitamin D suppresses glucocorticoid-induced transcription and cytotoxicity [181]. In the CNS, the most intense staining for vitamin D receptor and activating enzyme is described to be in the hypothalamus, including in the paraventricular nucleus (PVN) containing the corticotrophin releasing hormone (CRH)-positive neurons [173]. These neurons also stain positive for vitamin D 24-hydroxylase, and therefore are likely vitamin D responsive [182]. 

The anti-aging protein klotho plays a key role in regulating vitamin D metabolism. Membrane bound klotho is a cofactor for FGF23. Together, they form the receptor complex instrumental in Vitamin D production [183]. The biological functions of Vitamin D and klotho are highly intertwined because vitamin D induces the expression of klotho, and klotho keeps vitamin D levels in check. Klotho inhibits 1α-hydroxylase to decrease the active form of vitamin D—1-25(OH)2 D3, and increases activity of 24-hydroxylase, which converts both vitamin D and 1-25(OH)2 D3 into 24-hydroxylated products targeted for excretion. Lastly, low vitamin D levels are associated with depression and chronic stress, both conditions also linked to decreased klotho and elevated cortisol [60,184,185]. 

## 6. Cortisol and Klotho Associations with PD Symptomatology

### 6.1. Mood and Cognition 

Initial studies revealed that cortisol and klotho may influence non-motor symptoms of PD. Cortisol levels have most commonly been correlated with neuropsychiatric symptoms. A large number of people with Major Depressive Disorder (MDD) show abnormalities in the HPA axis functioning, with coinciding increased plasma levels of cortisol [185]. MDD patients also show neurochemical changes in CRH in the PVN, a structure now known to contain inclusions of α-synuclein, hallmark of PD pathology [186]. In individuals with PD, cortisol has been shown to correlate with the severity of depression [187] and with prevalence of anxiety and anhedonia [67]. In PD patients with impulse control disorders, increased cortisol is associated with more risk-taking behavior [188].

There is a growing body of evidence that increased cortisol is associated with late-life cognitive decline in normal aging in people with pre-clinical or clinical AD [57,189,190]. In non-demented patients, high cortisol correlates with decreased total brain volume, particularly in grey matter, and poorer cognitive function [191,192]. Elevated cortisol has also been linked to hippocampal atrophy, correlating with memory dysfunction [193]. Studies are needed to evaluate the link between cortisol and cognition in PD.

While klotho levels have not been related to psychiatric symptoms in PD itself, low klotho levels have been associated with depression [60]. Interestingly, a small exploratory study also found that CSF klotho levels are increased by electroconvulsive treatment for depression [194]. Lastly, *KL* gene polymorphisms can influence responsiveness to selective serotonin reuptake inhibitors (SSRIs) in people with depression [195]. 

Klotho has been shown to confer cognitive resilience in healthy aging and neurodegenerative disease. In animal models, klotho overexpression increases long-term potentiation and enhances spatial learning and memory [17]. In normal aging individuals, carrying the *KL* genetic variant (*KL*-VS), resulting in higher system klotho protein levels, links to enhanced cognition and enhanced functional brain connectivity [18,108]. Higher klotho levels also associate with enhanced volume of dorsolateral prefrontal cortex, an area that drives executive function [196]. Clinical studies studying klotho in relation to cognition in PD are lacking but initial studies reveal that acute elevation of klotho by peripheral delivery is sufficient to restore cognition in transgenic mouse models of PD [71]. 

### 6.2. Circadian Rhythm

In recent years, it has become increasingly apparent that the circadian rhythm influences PD, with patients experiencing diurnal fluctuations in motor and non-motor symptoms, despite stable pharmacokinetics of dopaminergic medications [197]. While mechanisms behind this remain unclear, it is known that neurodegeneration affects the central structures responsible for sleep and wakefulness, which may in turn affect input to the hypothalamic SCN, housing the molecular clock of the circadian system. This molecular clock consists of core clock genes, and disruption in their function in the pathogenesis of PD has recently gained attention [198]. In people with PD, degeneration of the dopamine containing cells in the retina may further affect input needed for alignment of dark/light cycles [199]. Lastly, dopaminergic therapy has a bidirectional relationship with circadian rhythm—responsiveness of motor symptoms to medication declines later in the day and medication leads to uncoupling of circadian and sleep regulation [200,201,202].

The circadian pattern of cortisol secretion provides a key signal from the SCN to peripheral clock genes. PD-associated changes in HPA axis function leads to a flatter circadian profile for cortisol and signaling to peripheral clock genes is compromised with resultant circadian disorder [65,198]. It is interesting to note that chronic kidney disease is associated with dysregulation of the SCN [203], and kidney disease is a risk factor for PD. 

Unlike the hormone cortisol, there is not much written about klotho and circadian function. An early report in healthy humans found that serum klotho showed a circadian rhythm with falling levels in the evening, a marked nadir at midnight and levels rising again by the morning [204]. No such circadian variation in klotho has been found in human CSF [36] or in the serum of healthy rats [205]. However, a relationship between klotho and sleep appears more robust with subjective sleep quality being positively associated with klotho in sedentary middle-aged adults [206]. Lower levels of klotho are also reported in people with obstructive sleep apnea and associated with overnight markers of hypoxemia [207]. Klotho levels are also decreased with excessive sleep duration [208], which is known to increase the risk of inflammatory diseases. 

### 6.3. Motor Symptoms

When looking at healthy aging populations, cortisol negatively correlates with grip strength [209] while lower levels of klotho associate with decreased grip strength and knee strength [87,117]. In people newly diagnosed with PD, higher cortisol levels correlate with greater motor burden of disease, measured using the-Unified Parkinson’s Disease Rating Scale (UPDRS) part III [210]. Thus far, only one study reports that lower CSF levels of klotho link to increased UPDRS III scores and higher Hoehn and Yahr disease stage [37]. When mice deficient in klotho were initially described, a parkinsonian phenotype was noted, with development of hypokinesis and decreased stride length along with midbrain dopaminergic neuronal loss at 5 weeks of age [30]. Later studies demonstrated that treatment with klotho, either via intracerebroventricular injection in toxin mouse model of PD or administered intraperitoneally in α-synuclein transgenic mouse model of PD, ameliorates motor deficits [71]. More studies are needed in humans to confirm that changes in cortisol and klotho affect PD motor symptoms. 

## 7. Proposed Model

Evidently there exists a complex interplay between aging and chronic stress in the pathogenesis of PD through mechanisms involving mitochondrial dysfunction and oxidative stress, neuroinflammation, insulin resistance, and vitamin D. It is difficult to extract precise cause and effect in the vicious circle resulting in neurodegeneration; however, there is a case that the hormones cortisol and klotho can contribute to the disease process in a yin-yang manner. Moreover, given their broad effects, these hormones may be key players in both idiopathic and familial PD. Our model proposed in Figure 1 suggests that in PD, there is accelerated aging and increased stress, leading to decrease in circulating klotho and increase in cortisol. As the balanced dualism of these hormones becomes dysregulated, there is resultant pro-inflammatory environment and mitochondrial dysfunction facilitating a wide range of pathways leading to neurotoxicity. Vitamin D metabolism may also be shifted along with increase in insulin resistance, further facilitating disease processes. Interestingly, both cortisol and klotho levels may be amenable to change, with physical activity increasing klotho and decreasing cortisol, and psychological stress/depression decreasing klotho and increasing cortisol.

## 8. Conclusions

In summary, we have highlighted the multifactorial actions of cortisol and klotho in the pathogenesis of PD. Downstream effects of cortisol and klotho may influence non-motor and motor symptoms of PD. Given that PD is a heterogenous disorder with multiple pathways involved in neurodegeneration, identifying strategies with a broad neuroprotective potential, e.g., via exercise-induced modifications of cortisol and klotho secretion, offers the potential of increasing the brain’s overall resilience. Future studies on how these hormones of aging and stress play a role in PD will lend evidence to whether these can be potential biomarkers or novel targets for interventional strategies.

## Figures and Tables

**Figure 1 brainsci-12-01695-f001:**
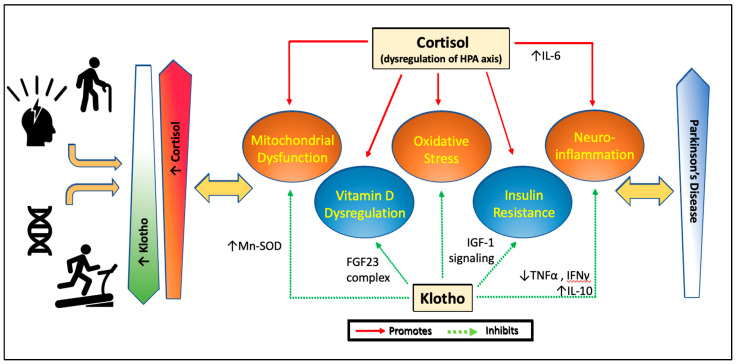
**Proposed model for effects of cortisol and klotho on pathways linked to Parkinson’s disease.** With stress and aging, cortisol levels increase due to dysregulation of the HPA axis and klotho levels decrease. Exercise decreases cortisol levels and increases klotho levels. Certain genetic variations may further dampen cortisol sensitivity and cellular response to stress or increase klotho levels or change its function. Chronic elevation of cortisol with aging or stress leads to increase in mitochondrial dysfunction, oxidative stress and activation of inflammatory cytokines, while promoting insulin resistance and correlating with lower vitamin D levels. Klotho normally protects mitochondrial structure and function, decreases oxidative stress, decreases inflammatory states, suppresses the downstream signaling pathway that leads to insulin resistance, and downregulates vitamin D metabolizing enzymes to control active vitamin D levels. This dysregulation of cortisol and reduction in klotho with aging and stress, important risk factors for PD, are hypothesized to affect the course of PD. Changes in cortisol and klotho in disease conditions may affects signs, symptoms, or progression of PD.

## Data Availability

Not applicable.

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
