# Peer review of "The Interrelated Multifactorial Actions of Cortisol and Klotho: Potential Implications in the Pathogenesis of Parkinson’s Disease"

_brainsci, 2022, doi:10.3390/brainsci12121695_

Round 1
Reviewer 1 Report
The current manuscript is interesting since it recognizes the significance of cortisol and klotho in the field of neurodegeneration.However, this review required a fewadditional details that would be useful to the audience.
1) The manuscript lacks the Genomic/chromosomal location of the klotho, which the author must add.
2) Author was required to demonstrate the correlation or connection between the key pathogenic PD genes and klotho (which is very minimally explained in the manuscript).
3) A more comprehensive link between motor symptoms and klotho is necessary, to demonstrate the significance of klotho in PD rather than highlighting other neurodegenerative diseases such as AD.
4) Author must explain the significance of klotho in idiopathic and familial Parkinson's disease.
5) The association of klotho with levodopa will benefit readers.

Reviewer 2 Report
This is a very interesting review on Pathogenesis of Parkinson’s Disease.
In this review article the authors evaluated the interdependent, opposing effects of cortisol and klotho in the etiology of Parkinson's disease. The authors stated that collectively, they influence powerful and distinct pathways that may influence PD-related symptoms. Better understanding of these hormones in Parkinson's disease might assist the design of effective therapies that can simultaneously influence the many systems involved in the pathogenesis of Parkinson's disease. The paper is well organized and written, however, there are few suggestions:
Minor concerns:
- The authors should increase the quality/size of figure 1 for better visualization.
- Line 51 “These pathways result in changes in cellular function……” authors should give few details about the pathways
- Line 51 114 “ Cortisol and klotho have relevance to healthy aging and various neuropsychiatric disorders ….” The authors should include few examples
